# A Rfa1-MN–based system reveals new factors involved in the rescue of broken replication forks

Ana Amiama-Roig ◉°, Marta Barrientos-Moreno ◉°, Esther Cruz-Zambrano, Luz M. López-Ruiz ◉, Román González-Prieto ◉, Gabriel Ríos-Orelogio ◉, Félix Prado ◉*

Centro Andaluz de Biología Molecular y Medicina Regenerativa (CABIMER), Consejo Superior de Investigaciones Científicas, Universidad de Sevilla, Universidad Pablo de Olavide, Seville, Spain

◉ These authors contributed equally to this work.
* felix.prado@cabimer.es

## Abstract

The integrity of the replication forks is essential for an accurate and timely completion of genome duplication. However, little is known about how cells deal with broken replication forks. We have generated in yeast a system based on a chimera of the largest subunit of the ssDNA binding complex RPA fused to the micrococcal nuclease (Rfa1-MN) to induce double-strand breaks (DSBs) at replication forks and searched for mutants affected in their repair. Our results show that the core homologous recombination (HR) proteins involved in the formation of the ssDNA/Rad51 filament are essential for the repair of DSBs at forks, whereas non-homologous end joining plays no role. Apart from the endonucleases Mus81 and Yen1, the repair process employs fork-associated HR factors, break-induced replication (BIR)-associated factors and replisome components involved in sister chromatid cohesion and fork stability, pointing to replication fork restart by BIR followed by fork restoration. Notably, we also found factors controlling the length of G1, suggesting that a minimal number of active origins facilitates the repair by converging forks. Our study has also revealed a requirement for checkpoint functions, including the synthesis of Dun1-mediated dNTPs. Finally, our screening revealed minimal impact from the loss of chromatin factors, suggesting that the partially disassembled nucleosome structure at the replication fork facilitates the accessibility of the repair machinery. In conclusion, this study provides an overview of the factors and mechanisms that cooperate to repair broken forks.

## Author summary

The cellular mechanisms that respond to broken replication forks remain poorly understood, despite the fact that genomic instability arising during DNA replication is a hallmark of early cancer progression. A major limitation in addressing this gap is the absence of robust systems to systematically screen for the genetic factors involved. Recently, genetic systems have been developed to induce replication fork breakage via a DNA nick—an intermediate step that is physiologically relevant in DNA repair and topological regulation. However, the cellular response to direct double-strand breaks (DSBs) at

**Data availability statement:** All relevant data are within the manuscript and its Supporting information files.

**Funding:** This work was supported by Ministerio de Ciencia e Innovación (MCIN/ AEI/10.13039/501100011033) and European Regional Development Fund (ERDF A way of making Europe) (grant PID2021-127486NB-100 to FP), and Ministerio de Ciencia e Innovación (MCIN/ AEI/10.13039/501100011033) and European Social Fund (ESF investing in your future) (pre-doctoral fellowship PRE2022-101266 to AA-R). The funders had no role in study design, data collection and analysis, decision to publish, or preparation of the manuscript.

**Competing interests:** The authors have declared that no competing interests exist.

replication forks, such as those resulting from fork collapse or unscheduled nuclease activity, remains largely unexplored. In this study, we engineered a chimeric protein, Rfa1-MN, which fuses the largest subunit of the single-stranded DNA-binding complex RPA with micrococcal nuclease (MN). This chimera preferentially generates DSBs at replication forks, enabling us to screen for mutants impaired in fork repair. Our screening identified novel factors that highlight the significance of error-prone break-induced replication (BIR) restart, fork restoration from BIR-intermediates and rescue by converging forks. Specifically, recombination factors associated with replication forks, replisome components critical for fork stability, and regulators of the G1 phase—controlling replication origin number— are potential players to regulate the efficiency of these pathways and the impact of broken fork repair on genome integrity.

## Introduction

DNA replication duplicates the genome during the S phase of the cell cycle. This essential process requires the coordinated firing of multiple replicons with bidirectional replisomes copying large genomic regions. The integrity of the replication fork is threat by its intrinsically fragile molecular nature (a dynamic nucleosome-free structure with DNA ends and single-stranded DNA; ssDNA) and the presence of multiple factors that hamper its advance (DNA adducts, abasic sites, ribonucleoside monophosphates (rNMPs), specific DNA structures like G-quadruplexes or R-loops, other processes like transcription and unbalanced supplies of deoxynucleoside triphosphates (dNTPs) or histones) [1]. Dealing with these situations is critical not only for a timely completion of genome duplication but also to prevent genetic instability. Accordingly, cells are endowed with different mechanisms that protect and repair stalled forks [2–5]. Much less is known, though, about the mechanisms that deal with double-strand breaks (DSBs) at forks despite DSBs at linear molecules are one of the most deleterious DNA lesions and their repair has been extensively studied from yeast to human [6–9]. The reason is that most of those studies took advantage of DNA sequence-specific endonucleases that allowed to follow the repair process; in contrast, DSBs at forks are spread along the genome at different positions in each cell, making difficult their analysis.

DSBs at forks have been proposed to be repaired by break-induced replication (BIR), a homologous recombination (HR) process in which the homology is restricted to one end; upon invasion of a homologous template, DNA synthesis can proceed for large genomic regions [10]. BIR, which has been extensively characterized in yeast, does not assemble a canonical fork; instead, it proceeds through a conservative DNA synthesis mechanism that is associated with a migrating bubble-like replication fork in which the Polδ subunit Pol32 becomes essential [11]. This structure is highly mutagenic and unstable, leading to multiple template-switching events and genome rearrangements that resemble those occurring in cancer genomes [12,13]. In terms of repair proteins, the most relevant difference with other DSB-induced HR events is that it can occur – though more inefficiently – in the absence of Rad51 [14]. In accordance with BIR acting upon DSBs at forks, mutants defective in replication-coupled nucleosome assembly accumulate broken forks that are rescued by a Rad52-dependent, Rad51-independent HR mechanism [15]. However, it is unknown if this requirement is specific of broken forks under conditions of altered chromatin.

A major handicap to associate BIR with broken forks is that the systems to study BIR follow the repair of a DNA sequence-specific endonuclease-induced DSB with a homologous sequence located on an ectopic region. As an alternative, genetic systems have been used in which an induced nick is converted into a DSB when encountered by a replication fork [16–23]. DNA

nicks are physiologically relevant because they are common intermediates of DNA repair and topological processes that are targeted in therapeutic treatments in cancer. A nick at the leading template causes a single-ended DSB (seDSB) that is rescued by an error-prone BIR-like process [17–20]; however, BIR-associated synthesis is limited by two compensatory mechanisms: cleavage by the Mus81 endonuclease to convert the D-loop into a canonical fork and arrival of a converging fork [17]. Actually, the arrival of a converging fork before BIR might explain the detection of double-ended DSB (deDSBs) at some nicks at the leading template [20–22]. A nick at the lagging strand also leads to a seDSB using nicked plasmids in *Xenopus* egg extracts [19], and accordingly, DNA nicks in both leading and lagging strand templates can trigger BIR [22]. However, nickase-induced nicks at the lagging template can be bypassed by the replisome in yeast and mammalian cells leaving a deDSB behind the fork [20–23]. This bypass, though, depends on the nickase and structure of the DNA nick [19,20,22], indicating that the fork can respond to the nick and/or the nickase without collapsing.

Replication forks can also collapse and break directly under genetic or environmental conditions that cause replicative stress as those occurring during tumour development [24]. An *in vitro* approach to this type of DNA lesions treated *Xenopus* egg extracts with ssDNA-specific endonucleases such as S1 or mung bean nuclease, which cut preferentially at the fork where ssDNA accumulates under unperturbed conditions. This study showed that the replisome is partially dismantled after fork breakage but fully re-established by a HR process that requires the nuclease activity of Mre11, Rad51 and the initial DNA synthesis activity of Polε [25].

In this study, we have developed an *in vivo* system that induces DSBs preferentially at the replication forks and searched for mutants defective in their repair. This genetic analysis demonstrated that the HR factors involved in the formation of the ssDNA/Rad51 nucleofilament are essential for the repair of DSBs at forks. In contrast to canonical DSBs at linear molecules, the repair of DSBs at forks is facilitated by fork-associated HR factors, BIR-associated factors, replisome components and a timely G1 phase. These results suggest that cells deal with seDSBs at broken forks by two mechanisms: BIR followed by fork restoration and rescue by converging replication forks, and reveal new player controlling their efficiency.

## Results

### The chimera Rfa1-MN provides a genetic system to study the repair of DSBs at replication forks

Chromatin endogenous cleavage (ChEC) provides a method to detect protein chromatin binding [26]. This assay is based in the expression of a chimera of the protein of interest with the micrococcal nuclease (MN), whose nucleolytic activity is activated with $Ca^{2+}$ ions. If the protein is bound to DNA, activation of the MN domain will induce a detectable cut (Fig 1A, left). Since the intracellular levels of $Ca^{2+}$ ions are low for MN activation, this assay requires cells to be permeabilized with digitonin followed by addition of $CaCl_2$ (S1A Fig). This assay has also been used to follow the binding of repair proteins to non-DSBs DNA lesions, as the ssDNA fragments generated by the encounter of replication forks with methyl methanesulfonate (MMS)-induced DNA adducts [27–33].

An example of this approach is the chimera Rfa1-MN, which contains the largest subunit of the ssDNA binding complex RPA. RPA is an essential complex involved in replication fork stability, DNA repair and checkpoint activation [34,35]. After treating permeabilized cells with $CaCl_2$ for different times, total DNA was extracted and run into an agarose gel. As expected, the extent of DNA digestion was exacerbated in cells treated with 0.005% MMS (S1A Fig); however, in contrast to other chimeras like Rad52-MN that requires 30 minutes in the absence of MMS [27], DNA digestion by Rfa1-MN was detected after 5 minutes of $CaCl_2$

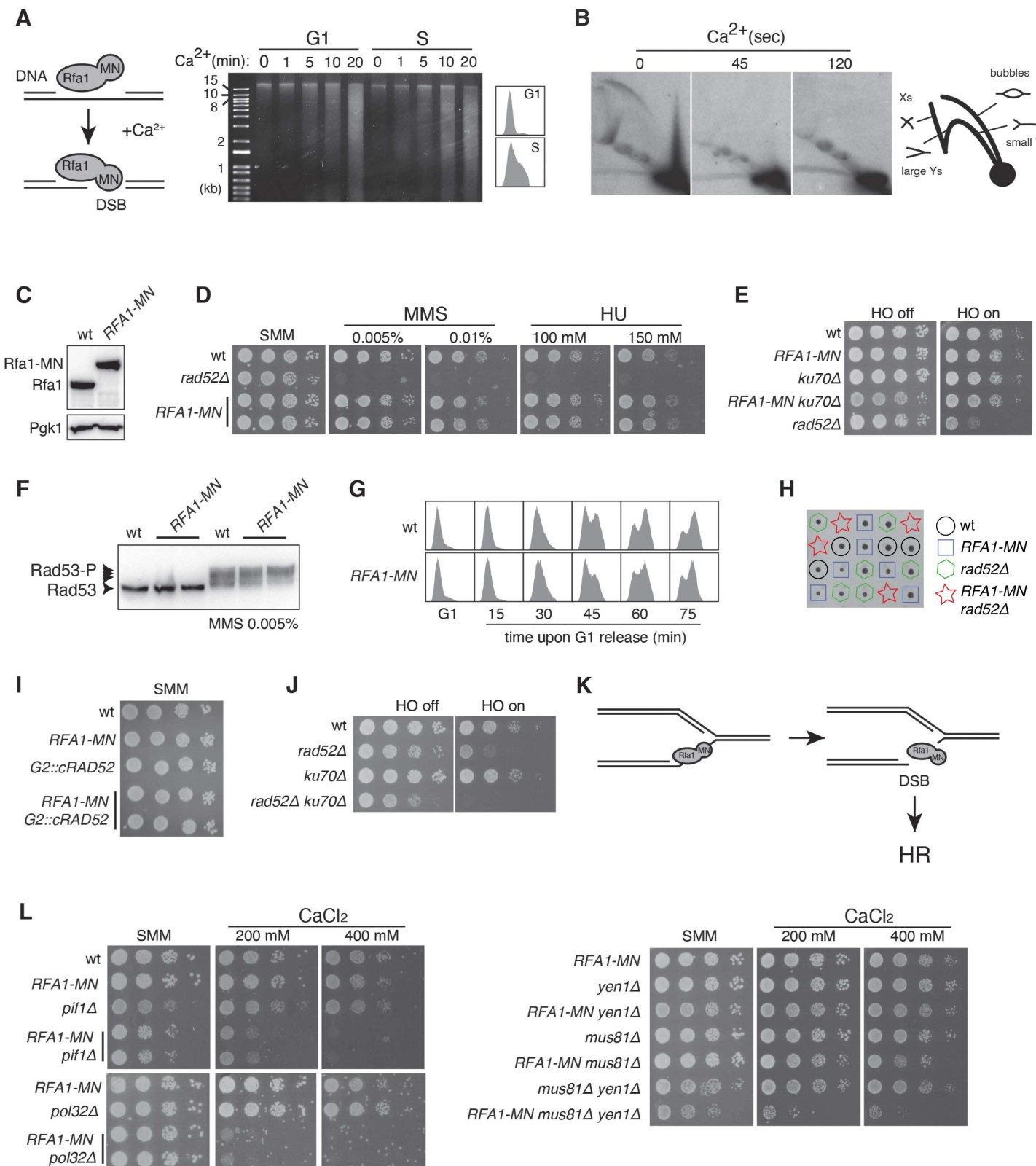

**Fig 1. The chimera Rfa1-MN provides a genetic system to study the repair of DSBs at replication forks. (A)** ChEC analysis of *RFA1-MN* cells arrested in G1 with α-factor and released into S phase for 30 minutes. Total DNA from cells permeabilized and treated with 2 mM CaCl₂ for different times is shown, as well as the FACS profiles. A scheme with the rational of the ChEC approach is shown on the left. **(B)** 2D/ChEC analysis of replication intermediates of *RFA1-MN* cells synchronised in

G1 with α-factor and released into S phase for 30 minutes. Total DNA from cells permeabilized and treated with Ca2+ for different times was digested with specific restriction enzymes and analysed by 2D electrophoresis. A schematic representation of the migration pattern of replication intermediates is shown on the right. **(C)** Rfa1 expression in wild-type and *RFA1-MN* cells from exponentially growing cultures as determined by western blot analysis. **(D)** MMS and HU sensitivity of *RFA1-MN* cells. Wild-type and *rad52Δ* cells were included as controls. **(E)** DSB sensitivity of *RFA1-MN* cells transformed with pGAL-HO and grown in glucose (*GAL1p* repression) and galactose-containing medium (*GAL1p* activation). An HO-induced DSB at the *MAT* locus can be repaired by NHEJ or, preferentially, by HR with the *HMR* or *HML* donor. The analysis was performed in wild-type and *ku70Δ* background (defective in NHEJ). **(F)** Rad53 activation in wild-type and *RFA1-MN* cells as determined by western blot analysis of exponentially growing cultures either in the absence or presence of 0.005% MMS for 1 hour. **(G)** Cell cycle progression of wild-type and *RFA1-MN* cells synchronised in G1 with α-factor and released into S phase for different times as determined by FACS analysis. **(H)** *RFA1-MN rad52Δ* lethality as determined by tetrad analysis. **(I)** Effect of restricting Rad52 expression to G2/M in wild type (*G2::cRAD52*) and *RFA1-MN* cells (*RFA1-MN G2::cRAD52*). **(J)** HO-induced DSB repair in cells defective in HR (*rad52Δ*) and/or NHEJ (*ku70Δ*). Cells were transformed with pGAL-HO and grown in glucose (*GAL1p* repression) and galactose-containing medium (*GAL1p* activation). **(K)** Proposed model for the essential role of HR in Rfa1-MN expressing cells. **(L)** Effect of the *pif1Δ*, *pol32Δ*, *yen1Δ*, *mus81Δ* and *mus81Δ yen1Δ* mutations in the growth of *RFA1-MN* cells in the absence and presence of different CaCl$_2$ concentrations. At high concentration, CaCl$_2$ can form crystals that did not affect the reproducibility of the assay. (D-E, I-J, **L**) Cell growth was determined by spotting 10-fold serial dilutions of the same number of mid-log growing cells onto the indicated mediums. All the analyses were repeated at least twice with similar results.

treatment. This digestion was observed both in cells maintained in G1 with α-factor and released into S phase for 30 minutes, although the kinetics of DNA digestion was faster in S phase cells (Fig 1A, compare 10 minutes digestion). This result is consistent with RPA localization at the transcribed regions of active genes [36]. However, RPA accumulates preferentially at replication forks where it protects ssDNA under unperturbed and stressed conditions, as determined by ChIP-seq of whole chromosomes and microscopy analyses of RPA foci in G1 and S phases [37–39]. Since ChEC preferentially detects DNA breaks at lineal molecules as the number of forks relative to the whole genome is low, we studied Rfa1-MN binding to replication forks by 2D/ChEC. In this assay, replication intermediates from ChEC-treated cells are analysed by 2-dimensional (2D) electrophoresis [28]. Activation of the MN activity of Rfa1-MN with Ca$^{2+}$ digested all replication intermediates in less than a minute (Fig 1B), in sharp contrast with other chimeras like Rad52-MN or Rad27-MN that required several minutes for a partial digestion [27]. Thus, although Rfa1-MN can induce DSBs at linear molecules, it preferentially digests replication forks.

The Rfa1-MN chimera is expressed at the same steady state level as the non-tagged protein (Fig 1C) and is proficient in DNA damage tolerance (Fig 1D), DSB repair (total and mediated by HR) (Figs 1E and S1B), checkpoint activation (Fig 1F) and DNA replication (Fig 1G). Only a slight delay from G1 to G2/M was observed by FACS, although the budding index and doubling time were similar in *RFA1-MN* and wild-type cells (S1C Fig). Importantly, the fact that the *RFA1-MN* mutant behaves as the wild-type strain in the presence of high concentrations of MMS and hydroxyurea (HU) indicates that the chimera is also proficient in replication fork processivity and stability even under high replication stress conditions.

Remarkably, the genetic combination *RFA1-MN rad52Δ* is lethal as determined by genetic analyses (Figs 1H and S1D). This synthetic lethality suggests that Rfa1-MN causes recombinogenic lesions that need to be repaired. HR deals with two different DNA lesions: DSB and replication associated-ssDNA. A major difference between them is that the former, but not the latter, can be repaired in *G2::cRAD52* cells that restrict the expression of Rad52 to G2/M [27]. We observed that the *RFA1-MN G2::cRAD52* strain grew as the wild type (Fig 1I), suggesting that Rfa1-MN causes DSBs. Rad52 essentiality in Rfa1-MN-expressing cells contrasts with the non-essential role for HR in the repair of mechanically- and HO endonuclease-induced DSBs where NHEJ can operate as a backup mechanism (Fig 1J) [40].

The simplest explanation to these results is that the amount of intracellular Ca$^{2+}$ is sufficient to induce the nucleolytic activity of Rfa1-MN at a rate that has no effect on cell growth unless HR is absent. Although we cannot discard the formation of some DSBs at other regions, the preferential accumulation of RPA at replication forks [37–39], the high efficiency

of Rfa1-MN to digest replication forks, and the essential role of HR for *RFA1-MN* cell viability suggest that most of these DSBs stem from Rfa1-MN–cut replication forks, preferentially at the lagging strand that accumulates most ssDNA/RPA (Fig 1K). We do not consider D-loops as a preferential target for Rfa1-MN-induced cleavage, because Rad52 does not promote, but instead prevents *RFA1-MN* cells lethality. Accordingly, *RFA1-MN* cells displayed a wild-type growth in the presence of high concentrations of HU even in plates enriched with $CaCl_2$ to increase the rate of cleavage (S1E Fig).

If most DSBs stem from digested replication forks, the expression of Rfa1-MN should display synthetic growth defects with mutations previously identified as required for the rescue of broken replication forks. We tested the absence of Rad51, Mre11, Pol32, Mus81 and Pif1 (Fig 1L). Just a few double mutants *RFA1-MN rad51Δ* and *RFA1-MN mre11Δ* germinated leading to microcolonies that grew better after streaking in a new plate, likely by adaptation to grow with lower level of intracellular calcium or the selection of suppressors (Figs 2A and S2A). The growth of the *RFA1-MN* strain was affected to different extents in the absence of Pif1, Pol32 and Mus81, but only in plates enriched with $CaCl_2$ (Fig 1L). The sensitivity of *mus81Δ* to nick-induced fork breakage is severely aggravated in the absence of Yen1 [17,21], an endonuclease that participates with Mus81 in the resolution of single Holiday junctions (HJs) [41] as those expected from the merging of a BIR-associated migrating D-loop with a converging fork. The absence of Mus81 and Yen1 strongly impaired cell growth in Rfa1-MN-expressing cells, although without being lethal (Fig 1L). In conclusion, Rfa1-MN provides a genetic system to search for factors involved in the repair of DSBs at replication forks.

## Identifying functions required for the repair of DSBs at replication forks

To search for factors required for the repair of DSBs at forks, we followed a Synthetic Genetic Array (SGA) analysis based in the crossing of an ordered array of null mutants to a strain harbouring the query allele *RFA1-MN* and specific markers such that the meiotic progeny with both the *RFA1-MN* allele and the null mutation can be scored for fitness [42]. This customized array encompasses 358 null mutants selected according to their confirmed or putative connection with the DNA damage response (S1 Table). The loss of viability or cell fitness was scored in plates without and with 400 mM $CaCl_2$ to increase the sensitivity of the screening. We obtained 62 hits, out of which 6 were wild type for the expected null mutation and 4 could not be validated by PCR (S2 Table). For manual inspection of these genetic interactions, we crossed the original *RFA1-MN* strain with each null mutant (including those scored as synthetically lethal), analysed genetically the dissected spores and studied the loss of fitness by drop assays in medium without and with different concentrations of $CaCl_2$. This study revealed 44 genes that are required to a greater or lesser extent for the viability of Rfa1-MN expressing cells (S2 Table). Except for *rad52Δ* and *pmr1Δ* mutants, we obtained double mutants with the *RFA1-MN* allele for the rest, including those scored as synthetically lethal in the SGA screening. These "lethal" mutants included *rad51Δ* and *mre11Δ* and displayed a similar behaviour (Figs 2A and S2A). Pmr1 is a $Ca^{2+}/Mn^{2+}$ ATPase required for $Ca^{2+}$ transport to Golgi whose null mutant accumulates excess $Ca^{2+}$ ions [43], which is likely causing a lethal number of broken forks.

## The repair of DSBs at replication forks requires DSB- and replication fork-specific HR activities

Apart from the mediator Rad52, we scored as synthetically lethal the MRX complex (Mre11, Rad50 and Xrs2), the recombinase Rad51 and its helpers Rad55 and Rad54 (Fig 2A and S2 Table), which are essential components of the HR machinery dealing with DSBs [44]. The

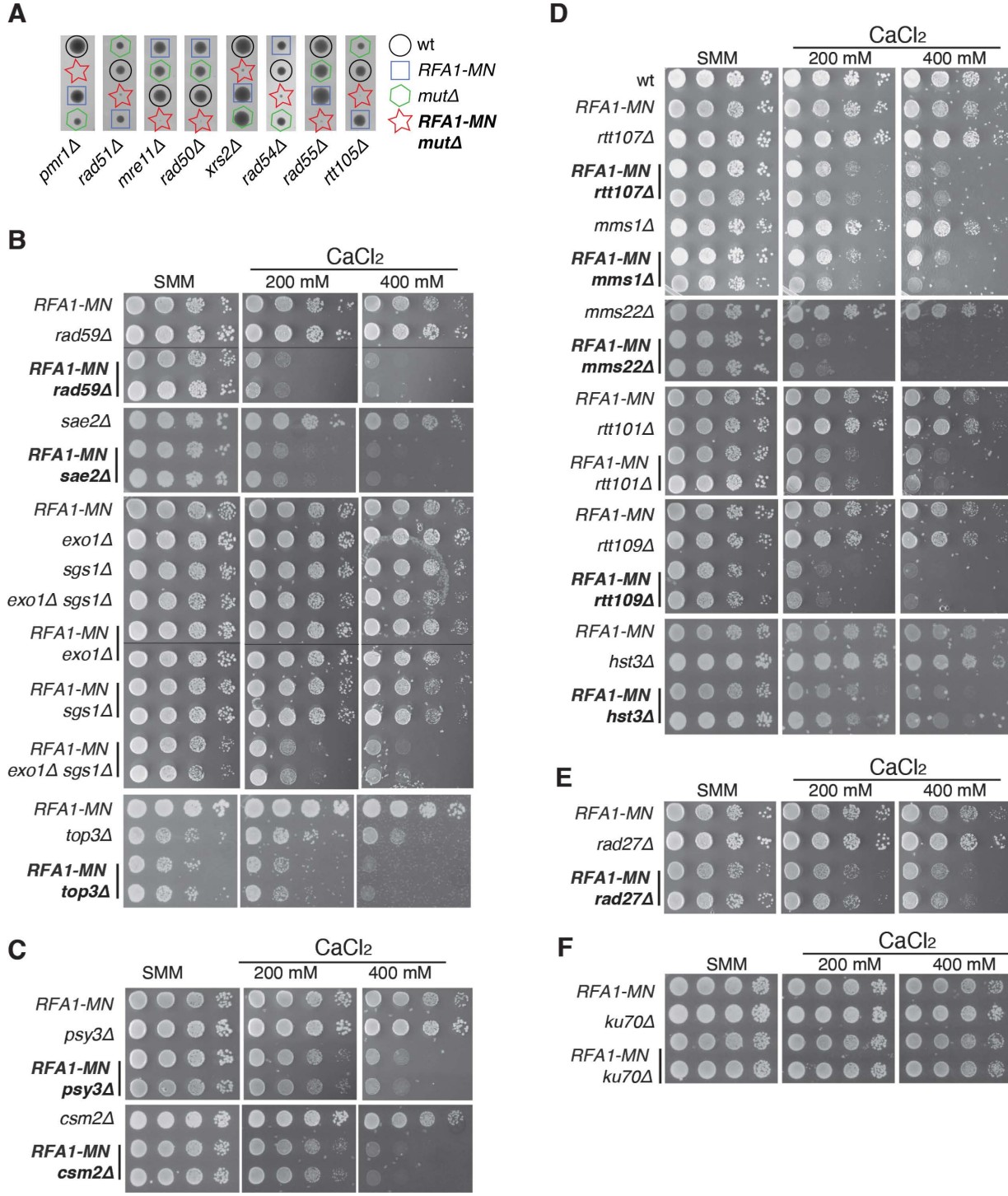

**Fig 2. The repair of DSBs at replication forks requires DSB- and replication fork-specific HR activities. (A)** Synthetic lethality of *RFA1-MN* with the indicated mutants as determined by tetrad analysis. **(B–F)** Effect of the indicated mutations in the growth of *RFA1-MN* cells as determined by spotting 10-fold serial dilutions of the same number of mid-log growing cells onto SMM medium without or with the indicated concentrations of CaCl$_2$. The analyses were repeated at least twice with similar results. Mutants scored in the SGA screening are shown in bold.

recombination protein Rad59 was also found in the screening but only in the presence of CaCl$_2$ (Fig 2B), which is consistent with the minor effect of *rad59Δ* in HR in the presence of Rad51 [45].

Another hit scored as synthetically lethal was Rtt105 (Fig 2A and S2 Table), despite it encodes a chaperone involved in the transfer to the nucleus and deposition at ssDNA of the RPA complex. However, the reduction in the level of RPA at forks in *rtt105Δ* cells is slight under normal conditions [37,46], which might explain why Rfa1-MN is inducing DNA damage as inferred from the lethality. The synthetic lethality of *RFA1-MN rtt105Δ* cells might be related to the function of Rtt105 in HR – where it facilitates Rad51 loading at ssDNA for DSB-induced gene conversion and BIR [46] – and to a lesser extent to the role of RPA in replication fork stability and checkpoint activation (see below). Actually, the DSB repair defect in *rtt105Δ* is almost as strong as that displayed by *rad52Δ* (S2B Fig). Remarkably, ssDNA stabilization by RPA is particularly critical for HR mechanisms that involves long-lived ssDNA intermediates, in particular BIR [47]. Altogether, these results demonstrate that the MRX complex, the Rad51/ssDNA nucleofilament and the factors that promote its assembly are essential for the repair of DSBs at forks.

During DSB-induced HR, Sae2 promotes the nuclease activity of the MRX complex in the initial processing of DSB ends to generate short stretches of 3'-ended ssDNA. This DNA resection is completed by the nuclease and helicase activities of Exo1 and Sgs1/Dna2 through complementary mechanisms [48]. In contrast to MRX, Sae2 was identified in the screening only in the presence of CaCl$_2$ (Fig 2B). To address the relevance of long resection, we analysed the effect on cell growth of the single and double *exo1Δ* and *sgs1Δ* mutants in combination with *RFA1-MN*. Only the *RFA1-MN exo1Δ sgs1Δ* displayed a loss of growth in the presence of CaCl$_2$ (Fig 2B). Again, the lack of long resection was not essential. It is worth noting the lack of effect of *sgs1Δ*, because the helicase Sgs1 is required for the dissolution of double HJ (dHJ) and sister-chromatin junction (SCJ) structures by the Top3/Sgs1/Rmi1 complex [49,50]. Remarkably, another hit of the screening was the topoisomerase Top3, scored as lethal (S2 Table). Since we could not validate the collection mutant by PCR, we generated the *RFA1-MN top3Δ* mutant by genetic cross. The lack of Top3 caused a growth defect in the presence of CaCl$_2$ (Fig 2B), suggesting a Sgs1-independent role in the rescue of broken forks.

Two components of the helper Shu complex (Psy3 and Csm2) were scored as synthetically sick in the presence of CaCl$_2$ (Fig 2C). The Shu complex is also involved in Rad51 filament formation [51], but in contrast to the aforementioned HR mutants, *shu* mutants are primarily sensitive to MMS-induced replication-associated ssDNA lesions but not to DSB-inducing agents [52,53].

Another functional genetic hub identified in the SGA screening is formed by *rtt109Δ*, *rtt107Δ*, *mms1Δ* and *mms22Δ* (Fig 2D) [54,55]. Rtt109 is a histone acetyltransferase that acetylates histone H3 at lysine 56 (H3K56) [56,57], which in turn facilitates histone H3/H4 deposition by increasing its interaction with chromatin assembly factors CAF and Rtt106 [58]. This pathway is stimulated after ubiquitylation of the acetylated histone by the Rtt-101$^{Mms22/Mms1}$ complex [59], which is associated with the replisome during S phase [60]. The involvement of the ubiquitin ligase Rtt101 was confirmed by manual inspection of *RFA1-MN rtt101Δ* mutants (Fig 2D). At chromatin, the H3K56ac/ Rtt101$^{Mms22/Mms1}$ pathway promotes the recombinational repair of replication-associated ssDNA lesions but not of DSBs [61–64]. The chromatin assembly and recombinational functions of H3K56ac can be separated in a double mutant *cac1Δ rtt106Δ* (Cac1 encodes the largest subunit of the CAF complex) because the ability of H3K56ac to stimulate nucleosome assembly depends on CAF and Rtt106 [58], whereas its ability to promote HR is independent of CAF and Rtt106 [65–67]. The triple *RFA1-MN cac1Δ rtt106Δ* was hardly affected even at high concentrations of CaCl$_2$ (S2C Fig),

suggesting that chromatin assembly does not play a major role in the repair of DSBs at forks. This was further confirmed by testing the *spt16-m* allele (alone or in combination with *cac1Δ rtt106Δ*), which impairs the replication-coupled nucleosome activity of the FACT complex [68], and the *pol1-2A2*, *mcm2-3A* and *dpb3Δ* alleles, defective in the transfer of parental histones to nascent strands [69–71]. Only the *RFA1-MN pol1-2A2* mutant displayed a weak effect in plates with high $CaCl_2$ concentrations (S2C Fig), which might be related to a subtle defect at its polymerase activity. The recombinational role of H3K56ac in the repair of DSBs at forks was further supported by the finding of the *hst3Δ* mutant in the screening (Fig 2D). Hst3 forms with Hst4 a Sirtuin complex that deacetylates chromatin-deposited H3K56ac once the replicative DNA damage is repaired [72,73]. Alternatively, the growth defect of *RFA1-MN hst3Δ* cells might be due to the inhibitory effect of H3K56 hyper-acetylation on DNA synthesis during BIR [74].

Rtt109 also facilitates the recruitment to stalled forks of Rtt107 through a H3K56ac-independent mechanism [75]. Rtt107 is a protein that acts as a scaffold for three genome maintenance complexes: the Rtt101[Mms22/Mms1] ubiquitin ligase, the Slx4 scaffold for the Slx1 and Mus81-Mms4 nucleases and the Smc5/6 SUMO ligase [76,77]. To address the role of the Smc5/6 SUMO ligase we tested the *smc6–56* allele and observed no effect on the growth of Rfa1-MN expressing cells (S2D Fig). This result is consistent with the dispensability of Sgs1 in broken fork repair, as the Smc5/6 complex is also required for MMS-induced SCJ dissolution and DSB repair [78]. We have further discarded a role for these structures in the repair of DSBs at forks by analysing the effect of a *rad18Δ* mutant, defective in PCNA ubiquitylation and replication stress-associated SCJ formation (S2D Fig) [79,80]. Therefore, the role of Rtt109 and Rtt107 on the growth of Rfa1-MN expressing cells is associated with the ubiquitylation and nuclease functions of Rtt101[Mms22/Mms1] and Mus81, respectively. In line with the latter function, it is particularly interesting the finding of Rad27 in the SGA screening (Fig 2E), as the physical and functional interactions of this endonuclease with Slx4-Mus81 (including synthetic lethality of the double null mutants) might be critical for the resolution of intermediates during replication stress [81]. Altogether, these results indicate that the repair of DSBs at forks requires replication fork-specific HR activities.

These results demonstrate that DSB- and replication-fork-associated HR functions participate in the repair of broken replication forks. On the contrary, NHEJ seems not to be required for broken fork repair because neither the Ku70/Ku/80 complex nor Nej1 were scored as positive hits. This was confirmed by manual inspection of a *RFA1-MN ku70Δ* mutant (Fig 2F).

## Checkpoint factors facilitate the repair of DSBs at forks

The second functional group involved in the repair of DSBs at forks encompasses several DNA damage checkpoint (DDC) (Rad9, the 9-1-1 complex (Ddc1/Mec3/Rad17) and its loader Rad24) (Fig 3A) and DNA replication checkpoint (DRC) factors (Mrc1, Tof1, and the Ctf8 and Dcc1 components of the PCNA loader RFC/Ctf18/Ctf8/Dcc1 (Ctf18-RFC complex)) (Fig 3B) [82,83]. All these factors have additional functions apart from checkpoint activation: Rad9 protects DSBs from premature resection [84]; the 9-1-1 complex participates in DDT and replication-coupled nucleosome assembly [85,86]; Mrc1 and Tof1 have roles in coupling helicase and polymerase activities, sister chromatid cohesion (SCC), and in the case of Tof1, stable fork pausing at replication fork blocks [87–91]; the Ctf18-RFC complex is required for replication fork stability upon stress and SCC [92,93]. In this case, since the annotated *ctf18Δ* mutant was wild type in the collection, we generated a new one and observed that it did not affect Rfa1-MN growth even at high $CaCl_2$ concentrations (Fig 3B). This result is consistent with previous observations showing that Ctf18, Ctf8 and Dcc1 are required for MMS and HU resistance, but only Ctf8 and Dcc1 are required for ionizing radiation (IR) and UV light

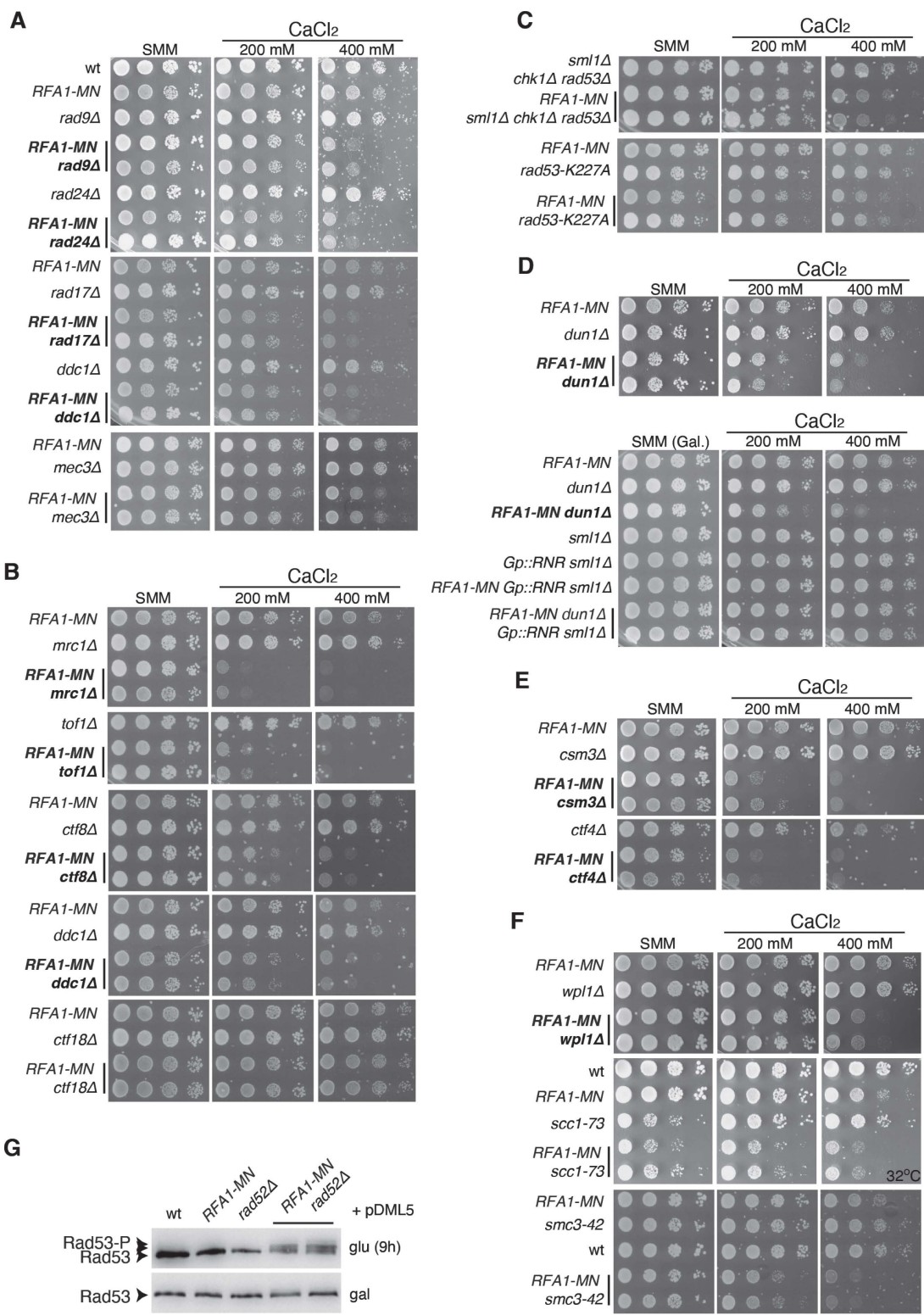

**Fig 3. Checkpoint, replication fork stability factors and cohesins facilitate the repair of DSBs at forks.** (A–F) Effect of the indicated mutations in the growth of *RFA1-MN* cells as determined by spotting 10-fold serial dilutions of the same number of mid-log growing cells onto SMM medium without or with the indicated concentrations of CaCl₂. Mutants scored in the SGA screening are shown in bold. (G) Checkpoint activation of the indicated strains transformed with plasmid pMDL5 (expressing

*RAD52* under control of the *GAL1* promoter) in galactose or after 9 hours in glucose, as determined by western blot against Rad53. The analyses were repeated at least twice with similar results.

resistance [94]. Since the whole complex is required for the aforementioned functions including DRC activation, these results suggest that Ctf8 and Ddc1 promotes broken fork repair by a not-yet defined function.

To test the effect of specifically eliminating the DNA damage and replication checkpoints, we generated a *RFA1-MN* strain lacking the checkpoint effectors Rad53 and Chk1 (*RFA1-MN sml1Δ rad53Δ chk1Δ*). This mutant displayed a subtle growth defect at high concentrations of $CaCl_2$ (Fig 3C). The kinase Dun1, target of Rad53, was also identified in the screening as synthetically sick in the presence of $CaCl_2$ (Fig 3D). A major role of Dun1 is to increase the levels of dNTPs during DNA replication and in response to DNA damage and replication stress [82,83]. According with this function, the calcium-induced defect is exacerbated in the presence of HU (S3A Fig), and more importantly, concomitant over-expression of the ribonucleotide reductase (RNR) complex that catalyses the rate-limiting step in dNTP synthesis and elimination of the RNR inhibitor Sml1 rescued the growth defects of the *RFA1-MN dun1Δ* mutant (Fig 3D). To determine if the partial checkpoint defect is exclusively due a deficit of dNTPs, we tested the kinase-deficient *rad53-K227A* mutant, which maintains wild-type dNTP levels [95]. This mutant also reduced the viability of Rfa1-MN-expressing cells at high concentrations of $CaCl_2$ (Fig 3C). The lack of effect of the checkpoint mutants under normal $Ca^{2+}$ conditions is consistent with the unphosphorylated state of Rad53 (Fig 1F). Unfortunately, we could not assess the checkpoint response to high $Ca^{2+}$ concentration because this condition caused a transient arrest in S phase in which the checkpoint was hardly activated (Fig S3B and C), and at later times (after 2–3 hours), $CaCl_2$ precipitates technically impeding western blot analysis. Therefore, we explored the checkpoint response under conditions of defective repair. Repression of *RAD52* in the *RFA1-MN rad52Δ* (pDML5) strain led to an accumulation of phosphorylated Rad53 (Fig 3G). This effect is partial, likely as a consequence of the basal expression of the *GAL1* promoter that allows *RFA1-MN GALp::RAD52* cells to slowly grow in glucose-containing medium (S1D Fig). Therefore, checkpoint activation facilitates cell growth under conditions that cause multiple broken replication forks, especially in HR defective cells.

## Replication fork stability factors and cohesins facilitate the repair of DSBs at replication forks

The lack of Mrc1 and Tof1 caused a severe growth defect in Rfa1-MN expressing cells as compared with the rest of checkpoint mutants (Fig 3A–C), suggesting a role for SCC and/or replication fork stability in the repair of DSBs at forks. Supporting this possibility, another hit of the SGA screening was Csm3 (Fig 3E), which together with Tof1 constitute the fork protection complex (FPC) that participate in both SCC and coupling of the helicase and polymerase activities at the fork upon replication stress [87–91]. The involvement of these processes is also supported by the finding of *ctf4Δ* in the SGA screening (Fig 3E). Ctf4 is a replisome component that physically bridges the helicase with the polymerase α (Polα) and other factors including Chl1 and Dia2, which in turn link DNA synthesis to SCC and fork assistance to stress, respectively [96–98]. Indeed, we have also found Dia2 in our screening, although a mutant lacking the Ctf4 interacting domain did not affect *RFA-MN* viability (S4A Fig). Additional roles of Ctf4 in MMS-induced SCJ and parental histone recycling are unlikely required for broken fork repair according to the lack of effect of the *rad18Δ* and chromatin assembly mutants, respectively (S2C and D Fig) [69,99].

A direct involvement of cohesin dynamics in the repair of broken forks is supported by the identification in the SGA screening of Wpl1 (Fig 3F), a factor needed for the removal of cohesive cohesins from chromatin [100]. To address if a defect in SCC can also impair the repair of broken forks, we tested two thermosensitive mutants affected in subunits of the cohesin complex (*scc1–73* and *smc3–42*) (Fig 3F) [101,102]. The addition of $CaCl_2$ to the medium improved the growth of the *scc1–73* mutant, suggesting an activation of stress chaperones. Importantly, the double mutants *RFA1-MN scc1–73* and *RFA1-MN scm3–42* displayed growth defects in the presence of $CaCl_2$. This indicates that cells need a sufficient pool of functional cohesins for the repair of DSBs at forks.

It is worth to note that only the absence of some SCC factors has an effect on the rescue of DSBs at forks. For instance, although Mrc1 and the Ctf18-RFC complex are involved in the Scc2/Scc4-dependent cohesin de novo loading, the defects of *ctf8Δ* and *dcc1Δ* on broken fork repair cannot be attributed to this function because it requires the whole complex [90]. Likewise, the conversion of cohesins at non-replicated DNA ahead of the fork into cohesive structures behind the fork requires Ctf4, Tof1/Csm3 and the helicase Chl1 [90], but the lack of the latter has no effect on the growth of Rfa1-MN expressing cells (S4B Fig).

## Shortening of G1 compromises the rescue of broken replication forks

The null mutants *sic1Δ* and *cdh1Δ* were scored as synthetically sick in the presence of $CaCl_2$, but only *sic1Δ* was confirmed in the drop test displaying a subtle effect at 400 mM $CaCl_2$ (Fig 4A and B). Sic1 and Cdh1 control the G1/S transition through complementary mechanisms; Sic1 is a cyclin-dependent kinase (CDK) inhibitor, whereas Cdh1 is an activator of the anaphase-promoting complex (APC) that promotes cyclin degradation [103]. In accordance with their overlapping functions, the double mutant *sic1Δ cdh1Δ* causes lethality due to a premature entry into S phase and insufficient number of licensed origins [104,105]. However, it is possible to get a G1 phase shorter than the one displayed by the single mutants in a *cdh1Δ* strain with *SIC1* under control of the *GAL1* promoter [104]. Under semi-permissive conditions, the expression of Rfa1-MN in a *cdh1Δ Gp::SIC1* strain caused severe growth defects and lethality in the absence and presence of 200 mM $CaCl_2$, respectively (Fig 4B). Likewise, the lack of Whi5, a transcriptional repressor of cell-cycle activators controlling the entry into S phase [106], reduced the growth of the *RFA1-MN sic1Δ* strain in the presence of $CaCl_2$ (Fig 4C). These results suggest that the shortening of the G1 phase compromises the rescue of broken forks, likely by reducing the number of licensed origins. Accordingly, slowing down the cell cycle by plating cells at 22 ℃ slightly rescued the growth defects of *RFA1-MN cdh1Δ Gp::SIC1* cells (Fig 4D).

## Discussion

In this study, we have generated a chimera of the largest subunit of the RPA complex with the MN that preferentially generates DSBs at replication forks and searched for mutants affected in their repair. Several major conclusions can be withdrawn. First, the core HR factors required for the detection and processing of DSBs to form a ssDNA/Rad51 filament are essential, as shown for nick-induced fork breakage [17,21]. HR might be operating at a 1) deDSB generated either between Okazaki fragments (Fig 5A) or 2) by collapse of a converging fork with the gap left at the non-broken strand (Fig 5B), or 3) at a seDSB generated after cleavage of the fork junction, more likely at the lagging strand (Fig 5C). In the latter case, a direct cleavage at the fork junction would likely disrupt the replisome structure thus preventing fork progression. The scarce formation of dHJ structures by double strand break repair (DSBR) suggested by the null effect of *sgs1Δ* and *smc5–56* mutants could be due to a low accumulation

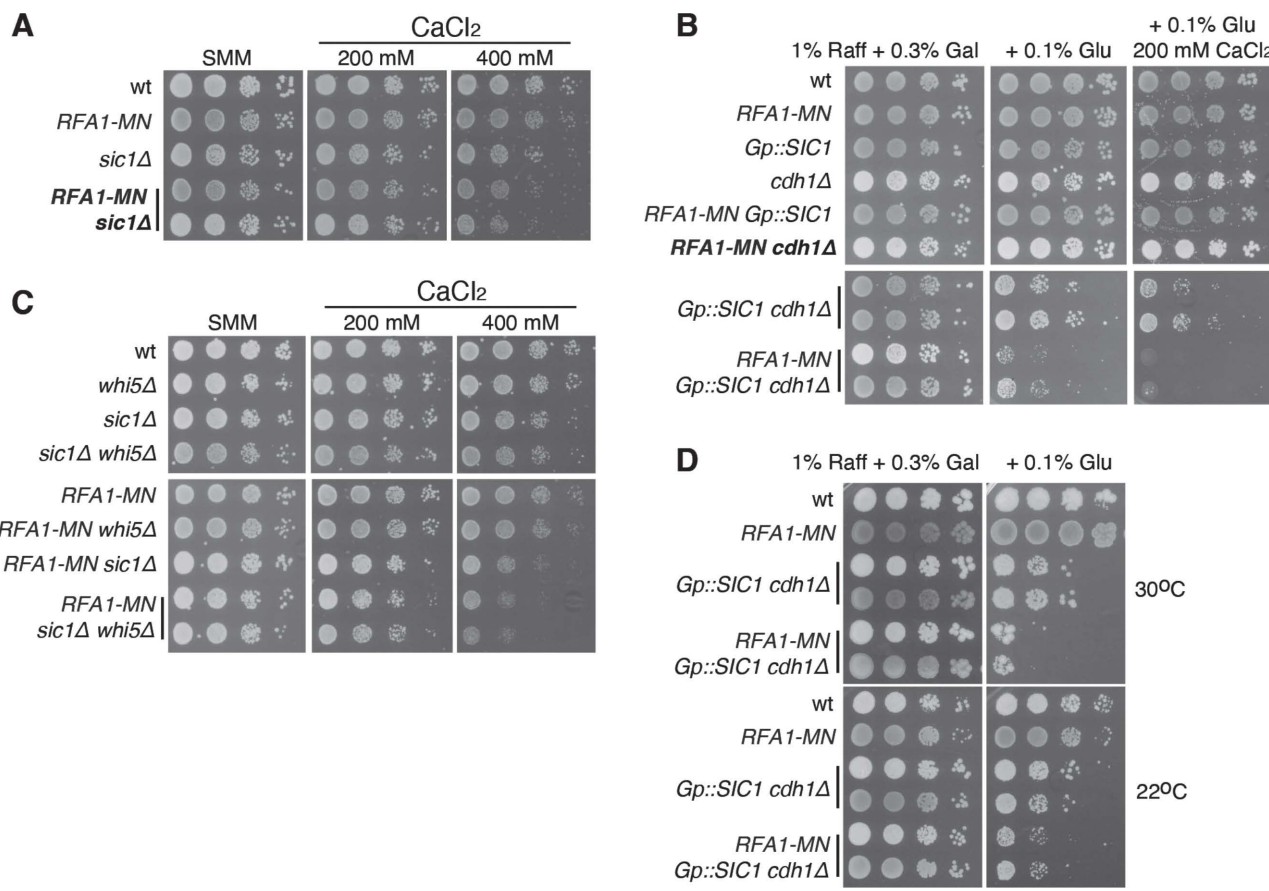

**Fig 4. G1 length facilitates the repair of DSBs at forks. (A–D)** Effect of mutations that shorten the length of G1 in the growth of *RFA1-MN* cells as determined by spotting 10-fold serial dilutions of the same number of mid-log growing cells onto SMM medium without or with the indicated concentrations of $CaCl_2$. To study a *sic1 cdh1* mutant (the double null mutant is lethal), we employed a strain where *SIC1* is under control of the *GAL1* promoter and analysed the effect of Rfa1-MN expression under permissive (0.3% galactose) and semi-permissive conditions (0.1% glucose) in the absence and presence of 200 mM $CaCl_2$. The analyses were repeated at least twice with similar results. Mutants scored in the SGA screening are shown in bold.

of deDSBs. Alternatively, it might reflect a preferential repair by synthesis-dependent strand annealing (SDSA), as observed for nick-induced deDSBs [22,23]. The accumulation of seDSBs by Rfa1-MN is supported by the finding of BIR-associated factors, fork-associated HR factors, and replisome components for their repair. These requirements are consistent with a BIR-like fork restart mechanism, even it is unlikely that extensive synthesis by BIR suffices for cell viability, as inferred from the growth defects of the triple mutant *RFA1-MN mus81Δ yen1Δ* (lacking the enzymes required for D-loop-to-fork conversion and HJ resolution after D-loop/fork merging) [17,21]. The growth defect of this mutant also reinforces the accumulation of seDSBs in *RFA1-MN* cells because Mus81 and Yen1 are not required for SDSA and are a backup mechanism for DSBR [8]. Since the arrival of a converging fork would lead to re-replication if the gap left at the non-broken strand has been repaired, seDSB invasion and D-loop formation could be a mechanism to ensure merging with the converging fork and genetic stability.

Our results also show that the DNA resection factors Sae2 and Sgs1/Exo1 are not essential, which might be explained by the minimal resection that replication-born seDSBs require for strand invasion [107] and the likely preferential cleavage of the lagging strand by Rfa1-MN

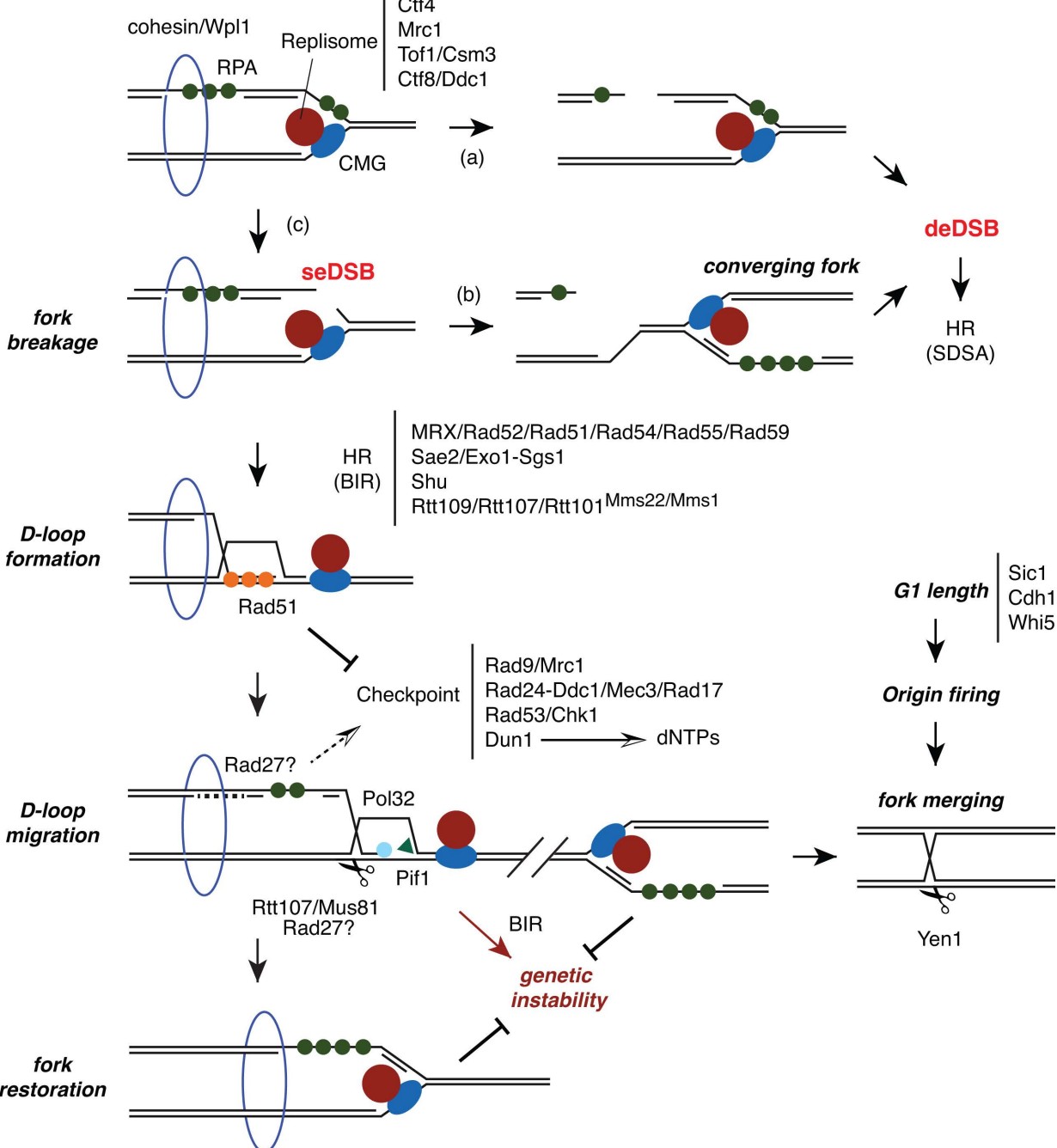

**Fig 5. Proposed mechanisms for the repair of a DSB at forks.** After fork breakage by Rfa1-MN, HR might be operating at a deDSB generated either between Okazaki fragments (**A**) or by collapse of a converging fork with the gap left at the non-broken strand (**B**), or at a seDSB generated after cleavage of the fork junction, more likely at the lagging strand (**C**). In response to seDSBs, the core HR machinery (with the help of fork-associated HR factors) would promote the invasion of the sister chromatid, generating a D-loop structure that primes a conservative, error-prone replication by a migrating bubble. This BIR-like restart mechanism would be facilitated by cohesins, checkpoint activation, Dun1-mediated increase in dNTPs, and replisome components that would be retained at the proximity of the D-loop for the stability of the migrating D-loop or, alternatively, the conversion of this structure into a canonical fork upon the activity of the Mus81 nuclease. The formation of the D-loop structure would prevent checkpoint activation and inhibition of late replication origins. The activation of these origins would also prevent BIR-associated genetic instability by fork merging with the D-loop structure and subsequent Mus81/Yen1-dependent HJ resolution. In line with this later mechanism, regulation of G1 length by Sic1, Cdh1 and Whi5 would facilitate the rescue of broken replication forks by ensuring a sufficient number of active origins, especially in response to massive fork breakage or fork breakage at specific regions like the end of chromosomes or common fragile sites.

that would generate a 3'-ended ssDNA. In this regard, the essentiality of the MRX complex, also observed for Mre11 but not for its nuclease activity in nick-induced fork breakage [21], is more likely related to its replication fork stability and sister chromatid tethering activities [108,109]. Sae2 and Sgs1/Exo1 might be preferentially required for those cases in which Rfa1-MN cuts at the leading strand, which would lead to the formation of blunt or 5'-ssDNA ends. Sae2 would facilitate the removal of the Ku complex from these intermediates, in accordance with the increase in nick-induced BIR events in yeast cells lacking Ku70 [23].

Finally, our screening has revealed a Sgs1-independent role for Top3 in the rescue of broken forks. Apart from its dHJ and SCJ dissolution activity in conjunction with Sgs1, the Top3/Rmi1 complex dissolves nascent Rad51-mediated D-loops *in vitro* that might explain the extreme growth defect and hyper-recombination phenotype of the *top3Δ* mutant [110]. This anti-recombinogenic activity is in apparent contradiction with the essential function of HR in the rescue of broken forks. However, this activity might be important to prevent template switching during BIR, a genotoxic event that can lead to chromosome rearrangements [13].

Second, the repair of DSBs at forks is facilitated by HR factors that are specific of stalled replication forks. In particular, it requires the Shu complex and the Rtt109/H3K56ac/Rtt-101$^{Mms22/Mms1}$ pathway, which promote the recombinational repair of replication-associated ssDNA lesions but not of DSBs [52,53,61–64]. The Shu complex facilitates the formation of the Rad51 filament during ssDNA gap filling by physically recruiting the Rad55/Rad57 heterodimer to stalled forks [111], whereas the Rtt109/H3K56ac/ Rtt101$^{Mms22/Mms1}$ pathway seems to uncouple the DNA polymerases from the CMG helicase to facilitate recombination [60,66]. These requirements suggest that the recombinational repair of broken forks occurs in the context of DNA-bound replication factors, which is supported by the finding of several replisome components in our screening. A concomitant study searching for factors involved in the repair of nick-induced DSBs uncovered a role for the Rtt109/H3K56ac/ Rtt101$^{Mms22/Mms1}$ pathway only when the nick is at the template for the leading strand [21]. Whereas a nick at the template for the lagging strand can be bypassed generating a deDSB behind the fork that is repaired by SDSA, a nick at the leading template can generate a seDSB. Thus, Rtt109/ H3K56ac/ Rtt101$^{Mms22/Mms1}$ might not be specifically required for a DSB at the lagging strand, but for the BIR-mediated restart of a seDSB. Accordingly, this pathway was identified – together with the core HR factors and the MRX complex – among the requirements for the repair of seDSBs generated by rNMP-induced nicks, regardless of their position at the leading or lagging strand [112].

Third, the rescue of DSBs at forks is associated with unstable replication intermediates, as inferred from the loss of viability of Rfa1-MN cells lacking Ctf4, Mrc1, Ctf8, Dcc1 or the Tof1/Csm3 FPC. These factors, like Pol32, are dispensable for unperturbed DNA replication. They participate in SCC, stable fork pausing and coupling of the helicase and polymerase activities at the fork upon replication stress [87–93,96,97,113]. The involvement of cohesins is supported by the finding of Wpl1 in the screening, and further demonstrated with specific thermosensitive alleles of the cohesin complex. This finding is expected as holding sister chromatids together by the cohesin complex is needed for the repair of both canonical DSBs and stalled replication forks [114,115]. Another replisome component partially required for *RFA1-MN* cell viability is Rad27, a nuclease that participates in the maturation of the Okazaki fragments [116]. The rescue of nick-induced seDSBs is associated with high rates of mutagenesis and template switching events [17]. BIR studies with ectopic HR systems suggest that mutagenesis stems from an accumulation of ssDNA at the lagging strand behind the migrating D-loop structure [117]. Efficient processing of this strand might be important to prevent excess ssDNA that would destabilise this replication intermediate. Alternatively, the physical and functional interactions of Rad27 endonuclease with Slx4-Mus81 might be critical for the

processing of the D-loop [81]. It is important to remark that the growth defects observed in the absence of replication factors are unlikely due to an accumulation of ssDNA and a higher probability of fork breakage by Rfa1-MN or to the additive effects of fork cleavage and replication stress because the *RFA1-MN* mutant behaves both with and without calcium as the wild-type strain even in the presence of high concentrations of MMS and HU that strongly impair cell growth (Figs 1D and S1E).

The establishment of cohesion is achieved through two partially complementary mechanisms: the conversion of cohesins associated with unreplicated DNA ahead of the fork into cohesive structures behind the fork (dependent on Ctf4, Tof1/Csm3 and Chl1) and the loading of nucleoplasmic cohesins onto fork-associated nascent DNA (dependent on the cohesin loader Scc2/Scc4 and the Ctf8-RFC complex) [90]. Our results show that the absence of Chl1 or Ctf18 does not impact the repair of DSBs at forks. Thus, the role of Ctf4, Mrc1, Ctf8, Dcc1 and the Tof1/Csm3 complex in broken fork repair cannot be explained just by a defect in SCC. Conservative replication associated with D-loop migration uncouples the leading and lagging strands [118]. In a canonical fork, they are coupled through physical interactions of Ctf4, Mrc1 and Tof1/Csm3 with the CMG helicase and the DNA polymerases Pol ε and Pol α [87,119]. A potential rearrangement of these interactions in the migrating D-loop structure might be related to the recombinational role of the Rtt109/H3K56ac/ Rtt101[Mms22/Mms1] pathway, as the sensitivity to replication stress of cells lacking this pathway can be suppressed by mutations in Ctf4, Mrc1, Dpb4 (Pol ε) or Mcm6 that uncouple the CMG helicase from the DNA polymerases [60,66]. A screening for factors involved in the rescue of oncogene-induced stressed forks uncovered, together with the BIR proteins Rad52 and PolD3 (human ortholog of Pol32), the FPC components Tipin and Timeless (human orthologs of Tof1 and Csm3) [120], suggesting a conservation of these factors.

Fourth, shortening of the G1 phase compromises the rescue of broken forks, as inferred by the inverted correlation between G1 length and cell growth defects in *RFA1-MN* cells lacking different inhibitors of the G1/S transition. Converging forks limit the mutagenicity associated with the repair of a nick-induced DSB, likely by merging with the D-loop [17]. Since a premature entry into S phase reduces the number of licensed origins [104,105], the severe growth defects of Rfa1-MN-expressing cells in combination with a shortening of G1 might be due to a reduction in the number of active forks that could rescue the broken forks. In yeast and cancer cells, premature entry into S phase by CDK deregulation in G1 causes a reduction in the number of active replication origins and genome instability. This instability has been proposed to result from a higher frequency of fork collapse and/or the entry into mitosis with incompletely replicated genomes [121]. Our result suggests that it may also arise from unrepaired broken forks and/or excess BIR-induced mutagenesis.

Fifth, the repair of DSBs at forks by HR is an efficient process. The lethality of the double mutant *RFA1-MN rad52Δ* suggests that at least one fork per cell cycle is cut by the chimera. However, *RFA1-MN* cells did not display growth defects and the checkpoint was not required except at high levels of $CaCl_2$ (consistent with an accumulation of DSBs and/or BIR-associated ssDNA) or in the absence of Rad52 (consistent with the accumulation of DNA resection-mediated ssDNA at broken forks when strand exchange is abolished [107]). Interestingly, we have found Dun1 in our screening and demonstrated that the growth defect is due to a reduction in the levels of dNTPs, in line with the Dun1-dependent increase in both dNTPs and mutagenesis observed during BIR [12].

Taking into account our results and previous studies, we propose the following model for the repair of seDSBs at forks (Fig 5). A Rad51/ssDNA nucleofilament formed at the broken nascent strand would invade the sister chromatid in the context of the replisome machinery with the help of stalled fork-associated HR factors, leading to the formation of a D-loop

structure. This invasion step has to occur behind the CMG helicase, which may be retained at the proximity together with replisome components for further restoration of the replication fork. These replisome components might be required for the stability of the migrating D-loop (whose advance would require Pol32 and Pif1) and/or the conversion of this structure into a canonical fork upon the activity of Mus81. Cohesins would also contribute to the stability of this structure and/or to the previous invasion step. Replication fork restart by this BIR-like mechanism is associated with high levels of mutagenesis and template switching events. This genetic instability would be potentially restricted by specific factors like Rad27 and Top3, the conversion of the D-loop into a canonical fork and the merging with a converging fork, favoured by the licensing of sufficient replication origins during G1 phase. In this context, the nucleases Mus81 and Yen1 might also be required for the resolution of the HJ structure generated after fork merging. A major observation of this study is the essential role of the HR machinery. We think that HR-mediated strand exchange would not only promote replication fork restart, but would also prevent inhibition of origin firing by checkpoint activation, as replication is required for the rescue by converging forks.

Apart from the positive hits, some of which requires further investigation to understand their connection with broken fork repair (S4C Fig), our screening revealed a scarce impact by the loss of chromatin factors. This is unexpected taking into account their relevance during DNA replication and DSB repair [122]. Mutants affecting the deposition of newly and parental histones during replication hardly affected the viability of Rfa1-MN-expressing cells. Likewise, histone chaperones that participate in replication-independent nucleosome exchange (HIR, Nap1, Chz1) and chromatin remodelling factors (INO80, SWR1, ISW1, ISW2, SWI/SNF and RSC) were negative hits in the screening, with the exception of Chd1 (S2 Table and S4D Fig). Although the involvement of chromatin in the repair of DSBs at forks requires a more detailed analysis, one possibility to explain its low impact is that the partially disassembled nucleosome structure at the advancing fork facilitates the accessibility of the repair machinery.

A limitation of our system is that many of the hits were identified by adding $CaCl_2$ to the medium to increase the number of broken forks. This sudden increase in cytosolic $Ca^{2+}$ triggers the reprogramming of $Ca^{2+}$ transporters to restore physiological levels [123,124]. Thus, we cannot rule out that some of the hits might be specific of this $Ca^{2+}$ stress context. Moreover, the cleavage likely occurs preferentially at the lagging strand, where RPA tends to accumulate. It will be interesting to determine the effect of the analysed mutants if the DSB occurs preferentially at the leading strand.

In summary, our results provide new genetic requirements for the repair of broken forks and highlight the significance of error-prone BIR restart, fork restoration from BIR-intermediates and rescue by converging forks. Specifically, recombination factors associated with replication forks, replisome components critical for fork stability, and regulators of the G1 phase may potentially control the efficiency of these pathways and the impact of broken fork repair on genome integrity, especially in regions with low density of active origins like the end of chromosomes and common fragile sites (CFS) in mammalian genomes [125,126], which relies on BIR-like mechanism: MiDAS (mitotic DNA synthesis) and ALT (alternative lengthening of telomeres) [127]. Future molecular experiments will be required to test the different scenarios inferred from our genetic analyses.

## Materials and methods

### Yeast strains, plasmids and growth conditions

All *Saccharomyces cerevisiae* strains used are haploid derived from BY4741 or W303. Yeast strains used in this study are listed in S3 Table. Most strains were generated by genetic crosses.

Tagged and deletion strains were constructed by a PCR-based strategy [128]. pDML5 is a *URA3*-based centromeric plasmid that expresses *RAD52* from the galactose-inducible *GAL1* promoter. pGAL-HO is a *URA3*-based multicopy plasmid expressing the endonuclease HO from the *GAL1* promoter [129]. Yeast cells were grown in supplemented minimal medium (SMM) at 30 °C except for liquid cultures supplemented with 400 mM $CaCl_2$, which was performed at 26 °C to reduce $Ca^{2+}$ precipitation. For G1 synchronization, cells were grown to mid-log phase and α-factor was added twice at 60 min intervals at either 1 (*BAR1* strains) or 0.5 μg/ml (*bar1Δ* strains). Then, cells were washed three times and released into fresh medium with 50 μg/ml pronase.

### Synthetic genetic array analysis

The synthetic genetic array analysis (SGA) was performed as reported with some modifications [42]. The query strains (*RFA1-MN::NAT* and control *trp1Δ::NAT*) were crossed with a customized array of null mutants using a manual replicator. The double mutants with *RFA1-MN* were scored as synthetically lethal or synthetically sick by comparing their growth with the double mutants with *trp1Δ::NAT* on the $SD_{MSG}$-His/Arg/Lys-canavanine-thialysine-G418-nourseothricin plates. To address the effect of $Ca^{2+}$, both sets of double mutants were first replica plated to SMM and then to SMM supplemented with 400 mM $CaCl_2$.

### DNA damage sensitivity

The sensitivity to Rfa1-MN expression, zeocin, MMS, HU and HO expression was determined by spotting ten-fold serial dilutions of the same number of mid-log growing cells onto SMM medium without or with $CaCl_2$, zeocin, MMS and HU, or onto glucose and galactose-containing medium (HO). For ionizing radiation sensitivity spotted cells were irradiated and then grown under unperturbed conditions. All analyses were repeated at least twice with similar results.

### Cell growth analyses

Cell cycle was followed by DNA content. DNA content analysis was performed by flow cytometry as reported previously [130]. Cells were fixed with 70% ethanol, washed with phosphate-buffered saline (PBS), incubated with 1 mg of RNaseA/ ml PBS, and stained with 5 μg/ml propidium iodide. Samples were sonicated to separate single cells and analyzed in a FACSCalibur flow cytometer. The budding index (percentage of cells with bud) was determined by counting 100 cells at each time point and replicate. The doubling time was calculated by measuring the $OD_{600}$ from exponentially growing cultures as previously described [131].

### *In vivo* ChEC and ChEC/2D analyses

Chromatin endogenous cleavage (ChEC) and ChEC/2D analyses of *RFA1-MN* cells were performed as reported [28]. Briefly, cells grown under the indicated conditions were arrested with sodium azide (0.1% final concentration). For cleavage induction, cells were permeabilized with digitonin and incubated with 2 mM $CaCl_2$ at 30 °C under gentle agitation. For ChEC analyses, total DNA was isolated and resolved into 0.8% TAE 1× agarose gels. To analyse replication intermediates (ChEC/2D), total DNA was extracted as detailed, digested with *Eco*RV and *Hin*dIII, resolved by neutral/neutral two-dimensional (2D)-gel electrophoresis, blotted to nylon membranes, and analysed by hybridization with the $^{32}$P-labelled probe Or. Signal was acquired in a Fuji FLA5100 with the ImageGauge analysis program.

## Western blot

Yeast protein extracts to analyse Rad53 phosphorylation and Rfa1/Rfa1-MN expression were prepared using the TCA protocol [132]. Protein samples were resolved by 8% SDS-PAGE, probed with antibodies against Rad53 (Abcam, ab104232), Rfa1 (Abcam, ab221198) or Pgk1 (Invitrogen, 22C5D8) and detected with a peroxidase-conjugate antibody. The immunoluminescent signal was generated with either the WesternBright ECL (Advansta) or the Clarity Western ECL Substrate (BioRad) kit, acquired in a ChemiDoc MP image system and quantified with the Image Lab software (Biorad).

## Supporting information

**S1 Fig. Characterization of *RFA1-MN* cells. (A)** ChEC analysis of exponentially growing cells expressing Rfa1-MN incubated in the absence or presence of 0.005% MMS for 2 h. Total DNA from cells permeabilized and treated with 2 mM $CaCl_2$ for different times is shown (left). Addition of $Ca^{2+}$ is required for detection of Rfa1-MN-digested DNA, as determined by running total DNA of wild-type and *RFA1-MN* cells growing in the absence or presence of 0.005% MMS for 2 h (right). **(B)** Ionizing radiation and zeocin sensitivity of *RFA1-MN* cells, as determined by spotting 10-fold serial dilutions of the same number of mid-log growing cells. Wild-type and *rad52Δ* cells were included as control. **(C)** Budding index and doubling time of wild-type and *RFA1-MN* cells. The mean and standard deviation of three (budding index) and two (doubling time) independent experiments are shown. **(D)** Effect of the *RFA1-MN* chimera in the viability of wild-type and *rad52Δ* cells transformed with the *URA3*-based plasmid pMDL5 expressing Rad52 from the *GAL1* promoter in the indicated media, as determined by spotting 10-fold serial dilutions of the same number of mid-log growing cells. The lethality of the *RFA1-MN rad52Δ* strain was rescued with the *URA3*-based plasmid pDML5, which expresses *RAD52* from the galactose-inducible *GAL1* promoter. This strain is able to grow, even though slowly, under glucose-repressing conditions; however, this is due to basal expression from the *GAL1* promoter, as indicated by the lack of growth in the presence of fluoroorotic acid (FOA) where only Ura⁻ cells are able to grow. **(E)** Effect of HU and calcium in the viability of *RFA1-MN* cells. The *rad52Δ* strain was included to show the requirement of HR for the repair of HU-induced DNA lesions. The analyses were repeated at least twice with similar results.
(EPS)

**S2 Fig. HR and chromatin assembly requirements for *RFA1-MN* cell viability in the absence and presence of calcium. (A, C, and D)** Effect of the indicated mutations in the growth of *RFA1-MN* cells as determined by spotting 10-fold serial dilutions of the same number of mid-log growing cells onto SMM medium without or with the indicated concentrations of $CaCl_2$. **(B)** DSB sensitivity of *rtt105Δ* cells to HO-induced DSBs, as determined by spotting 10-fold serial dilutions of the same number of mid-log growing cells. Cells were transformed with pGAL-HO and grown in glucose (*GAL1p* repression) and galactose-containing medium (*GAL1p* activation). Wild-type and *rad52Δ* cells were included as control. The analyses were repeated at least twice with similar results. Mutants scored in the SGA screening are shown in bold.
(EPS)

**S3 Fig. Checkpoint activation facilitates the repair of broken forks. (A)** Additive effect of *dun1Δ* and HU in the growth of *RFA1-MN* cells as determined by spotting 10-fold serial dilutions of the same number of mid-log growing cells onto SMM medium without or with the indicated concentrations of $CaCl_2$ and HU. **(B)** Cell cycle progression and budding index

of wild-type cells synchronised in G1 with α-factor and released into S phase in the presence of 400mM $CaCl_2$. The mean and standard deviation of three independent experiments are shown. **(C)** Rad53 activation in wild-type cells treated or not with 0.005% MMS for 2 hours in the absence and presence of 400mM $CaCl_2$.
(EPS)

**S4 Fig. Additional genetic requirements for *RFA1-MN* cell viability in the absence and presence of calcium. (A–D)** Effect of the indicated mutations in the growth of *RFA1-MN* cells as determined by spotting 10-fold serial dilutions of the same number of mid-log growing cells onto SMM medium without or with the indicated concentrations of $CaCl_2$. The analyses were repeated at least twice with similar results. Mutants scored in the SGA screening are shown in bold.
(EPS)

**S5 Fig. Raw data for figure panels.** Original blots for the indicated figure panels are shown.
(EPS)

**S1 Table. *Saccharomyces cerevisiae* genes studied in the SGA screening.** Genes analyzed in the customized library of null mutants are shown.
(XLSX)

**S2 Table. Positive hits from the SGA screening.** The name of the positive hits, their PCR validation and the effect of the null mutant on *RFA1-MN* viability are indicated.
(XLSX)

**S3 Table. *Saccharomyces cerevisiae* strains used in this study.** Strains, genotypes and references are indicated.
(DOCX)

**S4 Table. Raw data for figure plots.** Raw values to build budding index and doubling time plots are shown.
(XLSX)

## Acknowledgments

We thank Arturo Calzada, Ralph E. Wellinger, Mónica Segurado and Pedro San Segundo for various strains and reagents.

## Author contributions

**Conceptualization:** Félix Prado.

**Data curation:** Ana Amiama-Roig.

**Formal analysis:** Ana Amiama-Roig.

**Funding acquisition:** Félix Prado.

**Investigation:** Ana Amiama-Roig, Marta Barrientos-Moreno, Esther Cruz-Zambrano, Luz M López-Ruiz, Román González-Prieto, Gabriel Rios-Orelogio.

**Project administration:** Félix Prado.

**Resources:** Félix Prado.

**Supervision:** Félix Prado.

**Validation:** Ana Amiama-Roig, Marta Barrientos-Moreno, Esther Cruz-Zambrano, Luz M López-Ruiz, Román González-Prieto, Gabriel Rios-Orelogio.

**Visualization:** Ana Amiama-Roig, Marta Barrientos-Moreno, Félix Prado.

**Writing – original draft:** Félix Prado.

**Writing – review & editing:** Félix Prado.

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
