## [Decision Letter · Decision Letter 0]

7 Oct 2024

Dear Dr Prado,

Thank you very much for submitting your Research Article entitled 'A Rfa1-MN–based system reveals the factors that participate in the rescue of broken replication forks' to PLOS Genetics.

The manuscript was fully evaluated at the editorial level and by independent peer reviewers. The reviewers appreciated the attention to an important problem, but raised some substantial concerns about the current manuscript related to interpretations of results and suggesting additional experiments which may strengthen points of the the manuscript. Based on the reviews, we will not be able to accept this version of the manuscript, but we would be willing to review a much-revised version. We cannot, of course, promise publication at that time.

If you decide to revise the manuscript for further consideration at PLOS Genetics, please aim to resubmit within the next 60 days, unless it will take extra time to address the concerns of the reviewers, in which case we would appreciate an expected resubmission date by email to plosgenetics@plos.org.

If present, accompanying reviewer attachments are included with this email; please notify the journal office if any appear to be missing. They will also be available for download from the link below. You can use this link to log into the system when you are ready to submit a revised version, having first consulted our Submission Checklist .

PLOS has incorporated Similarity Check , powered by iThenticate, into its journal-wide submission system in order to screen submitted content for originality before publication. Each PLOS journal undertakes screening on a proportion of submitted articles. You will be contacted if needed following the screening process.

To resubmit, log into your Editorial Manager account and select the option 'Revise Submission' in the 'Submissions Needing Revision' folder.

We are sorry that we cannot be more positive about your manuscript at this stage. Please do not hesitate to contact us if you have any concerns or questions.

Yours sincerely,

Dmitry A. Gordenin, Ph.D.

Academic Editor

PLOS Genetics

Geraldine Butler

Section Editor

PLOS Genetics

Reviewer's Responses to Questions

**Comments to the Authors:**

Reviewer #1: A Rfa1-MN–based system reveals the factors that participate in the rescue of broken replication forks

Broken replication forks are a major source of genomic instability, and the repair of broken forks is incompletely understood. Using budding yeast as a model system, this manuscript reports the use of a fusion protein between the large subunit of the single-stranded DNA binding protein RPA and micrococcal nuclease (MN) termed RFA1-MN. This genetic tool induces double-stranded DNA breaks near RPA bound to DNA upon the addition of calcium, which activates MN. Remarkably the fusion protein does not elicit MMS or HU sensitivity in otherwise wild type cells. The study reports a systematic screen with custom array of 358 gene deletion and identify some expected hits (HR, BIR proteins), as well as some unexpected hits (replisome factors, replisome stability factors, cohesion factors). Also of interest are the negative results as well as well the limited genetic interactions with chromatin components. Another interesting result concerns DDR signaling, in that Rad53 is not activated but Dun1 shows genetic interaction with RFA1-MN. This is unexpected from the canonical Mec1-Rad53-Dun1 pathway concept. Finally, the screen produced evidence that shortening the G1 cell cycle phase has an impact on RFA1-MN sensitivity.

Overall, the data are well presented and the conclusions generally well supported by the results. The hits in the screen have been validated, and some unexpected results were followed up with additional genetic experiments to support the conclusions. This screen opens many doors for follow-up and mechanistic analysis, producing a comprehensive view on processes and factors affecting repair of broken replication forks.

Major comments:

1) The premise appears to be that Rpa1-MN only cleaves replication forks. What is the evidence? There are other potential sources of ssDNA bound by RPA from other nuclear processes such as BER, NER, MMR, and transcription (R-loops). There should be some discussion of this point.

The result that G2::cRAD52 RFA1-MN grows like wild type is of interest that may suggest that the damage is S-phase related, addressing potentially to above-mentioned point.

To strengthen the conclusion, a comparison with G1::cRAD52 and S::cRAD52 alongside Rfa1-MN could be considered.

2) In Figure 1A, there is some evidence of cleavage independent of exogenously added calcium, which is augmented in the presence of MMS. This should be discussed.

3) It is stated on page 7 (near top) that that RPA is primarily at the fork. What is the evidence for this, and please give references.

4) I was surprised that the MRN complex along with Sae2 is needed her, as it is not required in frank DSB repair unless the DSB ends are modified or covalently bound by a protein. Does this suggest that the ends or not free or otherwise modified that requires processing for Rad51 filament assembly? This aspect should be discussed.

5) The effect of Rtt105 is curious, as it would been expected to suppress a negative effect of RFA1-MN, as it is a chaperone for RPA to bind ssDNA. This suggests that Rtt105 has other client proteins that are relevant here. This should be discussed more in detail.

6) The requirement for Dun1 in the absence of Rad53 activation is curious and unexpected from the Mec1-Rad53-Dun1 pathway concept. There are other pieces of evidence for Rad53-independent activation of Dun1 that should be discussed here. These results could help illuminate this non-canonical signaling axis. See for example PMID: 12556502 and discussion therein of other cases.

7) The result that a short G1 phase has a negative consequence in RFA1-MN cells may suggest that converging forks play a role, as discussed in the manuscript. Does a slower cell cycle, for example growth on minimal media, rescue this defect, as is observed in bacteria in mutants with replication issues?

Additional points:

8) I know there is a discussion about this, but I thoguth it was settled that Sae2 itself is not a nuclease but a cofactor of MRN.

9) Rad27 is not structure specific but cleaves multiple substrates, see PMID: 10567561 for example. This may affect the discussion of potential mechanisms involved.

10) BIR is only Rad51-independent in one special chromosomal context, and Pol32 cannot mechanistically substitute its role in homology search and strand invasion. The argument at the beginning of the discussion should be eliminated.

11) There is a reference on page 20 to Figure S4B, but there is no such figure. Is Table S2 meant here. Please correct.

12) FigS1A: There is a slight G1-S or S cell cycle delay in the Rfa1-MN construct. Please comment and discuss.

13) FigS2A: RFA1-MN is combined with several factors involved in the chromatin assembly mechanism leading to the conclusion that the pol1-2A2 mutant is the only one that displayed a weak effect on plates with high CaCl2 concentrations, which might be related to a subtle defect in its polymerase activity. It is suggested that the chromatin assembly mechanism does not play a major role in the repair of DSBs at forks. To provide a more comprehensive analysis, the RFA1-MN dpb3∆ strain should be analyzed as well.

14) Fig3A: One of the replicates of the Rfa1-MN mec3∆ strain showed no reduced viability compared to the Rfa1-MN and mec3∆ single mutants, while the other replicate exhibited reduced viability. Please comment on this inconsistency.

15) Please correct typo on page 14: change ‘Chd1’ to ‘Cdh1’.

Reviewer #2: Uploaded as an attachment

Reviewer #3: The review is uploaded as an attachment

**Have all data underlying the figures and results presented in the manuscript been provided?**

Reviewer #1: Yes

Reviewer #2: Yes

Reviewer #3: None

PLOS authors have the option to publish the peer review history of their article (what does this mean? ). If published, this will include your full peer review and any attached files.

**Do you want your identity to be public for this peer review?** For information about this choice, including consent withdrawal, please see our Privacy Policy .

Reviewer #1: No

Reviewer #2: No

Reviewer #3: No

---

## [Decision Letter · Decision Letter 1]

27 Feb 2025

PGENETICS-D-24-00968R1

A Rfa1-MN–based system reveals new factors involved in the rescue of broken replication forks

PLOS Genetics

Dear Dr. Prado,

Thank you for submitting your manuscript to PLOS Genetics. After careful consideration, we feel that it has merit but does not fully meet PLOS Genetics's publication criteria as it currently stands. Therefore, we invite you to submit a revised version of the manuscript that addresses the points raised during the review process.

Please submit your revised manuscript within 30 days Mar 29 2025 11:59PM. If you will need more time than this to complete your revisions, please reply to this message or contact the journal office at plosgenetics@plos.org. Please include the following items when submitting your revised manuscript:

We look forward to receiving your revised manuscript.

Kind regards,

Dmitry A. Gordenin, Ph.D.

Academic Editor

PLOS Genetics

Geraldine Butler

Section Editor

PLOS Genetics

Aimée Dudley

Editor-in-Chief

PLOS Genetics

Anne Goriely

Editor-in-Chief

PLOS Genetics

**Reviewers' comments:**

Reviewer's Responses to Questions

Reviewer #1: Amiama-Roig et al.

Revised manuscript

In the revision the authors did a nice job to address my comments with new experimentation, clarifications, and text changes. The consideration of and experimental evidence for non-replication fork targets of Rfa1-MN is a good addition. I do agree that the likely major target will be replication forks and postreplication gaps, as supported by the new experiment shown in Figure 4d showing some suppression at slower growth.

In their model and the discussion, the authors may want to consider the D-loop as an additional target. There is good evidence that RPA binds the displaced strand of the D-loop to stabilize. However, the data show lethality with an HR defect and no HR-dependent lethality, suggesting that D-loop cleavage is not an issue. Maybe this is a point to make in the discussion?

Reviewer #2: The authors have adequately addressed my comments, and the revised manuscript is much improved. There are still a few points that could be clarified by revisions to the text:

1. Page 8, line 1: I suggest changing to “..is expressed at the same steady state level as..”

2. Page 10: The rad59 mutation does confer DNA damage sensitivity in RAD51 cells indicating that its activity is not specific to RAD51-deficient cells.

3. The rtt105 mutant has a much less severe phenotype in assays for DSB-induced recombination than canonical HR mutants, such as rad51 (Wang et al., 2021; 2024). Thus, it seems unlikely that loss of the Rad51 mediator function is responsible for the rtt105 RFA1-MN lethality. I wonder if an additional/alternative explanation for the RFA1-MN rtt105 lethality is that there are more MNase-induced breaks in these cells. In the rtt105 background, there would be sub-saturating levels of RPA on ssDNA leaving more ssDNA exposed for MNase cleavage. In the presence of Rtt105, ssDNA would be saturated by RPA shielding the ssDNA from MNase cleavage.

4. Page 18: The lack of a requirement for NHEJ to repair damage caused by RFA1-MN does not imply seDSB formation. Recent papers using nCas9 and Flp-nick showed no contribution of NHEJ to repair, even for nicks on the lagging-strand template that give rise to deDSBs. It seems more likely that the ends of replication-coupled deDSBs are not amenable to NHEJ. Since unwinding by CMG would be required to generate ssDNA on the lagging strand template for RPA to bind, it is not clear to me how seDSBs could form using this system. Once CMG has separated the parental strands it should continue translocation on the leading strand template. In vitro studies from the Walter lab showed that CMG can translocate onto the parental duplex at a pre-existing nick on the lagging strand template to make a seDSB but that requires the nick to be ahead of CMG. Since there could be 100s of DSBs in the presence of Ca ions, the sensitivity of some mutants, such as mus81 yen1, could be due to a fraction of these DSBs being processed by DSBR instead of SDSA.

**Have all data underlying the figures and results presented in the manuscript been provided?**

Reviewer #1: Yes

Reviewer #2: Yes

PLOS authors have the option to publish the peer review history of their article (what does this mean? ). If published, this will include your full peer review and any attached files.

**Do you want your identity to be public for this peer review?** For information about this choice, including consent withdrawal, please see our Privacy Policy .

Reviewer #1: No

Reviewer #2: No

**Figure resubmission:**
---

## [Editor Report · Decision Letter 2]

10 Mar 2025

Dear Dr Prado,

We are pleased to inform you that your manuscript entitled "A Rfa1-MN–based system reveals new factors involved in the rescue of broken replication forks" has been editorially accepted for publication in PLOS Genetics. Congratulations!

Yours sincerely,

Dmitry A. Gordenin, Ph.D.

Academic Editor

PLOS Genetics

Geraldine Butler

Section Editor

PLOS Genetics

Aimée Dudley

Editor-in-Chief

PLOS Genetics

Anne Goriely

Editor-in-Chief

PLOS Genetics

Comments from the reviewers (if applicable):

**Data Deposition**

http://datadryad.org/submit?journalID=pgenetics&manu=PGENETICS-D-24-00968R2

**Press Queries**

---

## [Editor Report · Acceptance letter]

PGENETICS-D-24-00968R2

A Rfa1-MN–based system reveals new factors involved in the rescue of broken replication forks

Dear Dr Prado,

We are pleased to inform you that your manuscript entitled "A Rfa1-MN–based system reveals new factors involved in the rescue of broken replication forks" has been formally accepted for publication in PLOS Genetics! Your manuscript is now with our production department and you will be notified of the publication date in due course.

With kind regards,

Dorothy Lannert

PLOS Genetics

On behalf of:
